# AMPA Receptor Auxiliary Proteins of the CKAMP Family

**DOI:** 10.3390/ijms20061460

**Published:** 2019-03-22

**Authors:** Jakob von Engelhardt

**Affiliations:** Institute of Pathophysiology, University Medical Center of the Johannes Gutenberg University Mainz, 55131 Mainz, Germany; engelhardt@uni-mainz.de; Tel.: +49-6131-3925005

**Keywords:** AMPA receptor, glutamate receptor, auxiliary subunit, synaptic function, short-term plasticity, long-term plasticity, hippocampus, lateral geniculate nucleus

## Abstract

α-amino-3-hydroxy-5-methyl-4-isoxazolepropionic acid (AMPA) receptors are assembled of four core subunits and several additional interacting proteins. Cystine-knot AMPA receptor-modulating proteins (CKAMPs) constitute a family of four proteins that influence the trafficking, subcellular localization and function of AMPA receptors. The four CKAMP family members CKAMP39/shisa8, CKAMP44/shisa9, CKAMP52/shisa6 and CKAMP59/shisa7 differ in their expression profile and their modulatory influence on AMPA receptor function. In this review, I report about recent findings on the differential roles of CKAMP family members.

## 1. Introduction

AMPA receptors mediate most of the fast-excitatory transmission in the central nervous system of vertebrates. They are tetramers comprising combinations of two different types of the four subunits GluA1-GluA4 [1]. The subunit composition determines the functional properties of AMPA receptors such as Ca^2+^-permeability, deactivation, desensitization and recovery from desensitization. However, these functional properties also depend to a large extent on additional interacting proteins, of which more than 30 were identified in proteomic studies [2,3,4,5]. The vast majority of native AMPA receptor complexes in the brain contain most likely one or several interacting proteins that bind to the GluA-subunits. Some of the interacting proteins are only part of the intracellular AMPA receptor complexes. These proteins influence the assembly, endoplasmic reticulum (ER) export and trafficking of AMPA receptors via the Golgi complex to the cell surface [6]. Furthermore, other proteins, which remain an integral part of the AMPA receptor complexes on the cell surface, influence receptor localization (synaptic or extrasynaptic) and function. These proteins are also known as AMPA receptor auxiliary proteins [7,8]. The group of AMPA receptor auxiliary proteins comprises the transmembrane AMPA receptor regulatory proteins (TARPs), cornichon homologs (CNIH-2 and CNIH-3), and GSG1L [2,3,4,9,10,11]. Several years ago, we identified CKAMP44/shisa9 as a novel AMPA receptor auxiliary protein [5]. A database search revealed that there are three genes with a high degree of similarity to CKAMP44 [12]. These genes encode for three proteins, namely CKAMP39/shisa8, CKAMP52/shisa6 and CKAMP59/shisa7, which also interact with AMPA receptors and modulate their expression and/or function [12,13,14,15]. CKAMP52, CKAMP59 and CKAMP44 therefore constitute a novel family of AMPA receptor auxiliary proteins. Although CKAMP39 can act at the AMPAR in non-neuronal cells, and thus may well be an auxiliary protein to AMPARs, this awaits further experimental evidence in the brain. CKAMP family members differ, similarly to the TARP family members, in their regional expression profile and modulation exerted on AMPA receptor function. In this review, I provide an overview on the current knowledge of CKAMP39, CKAMP44, CKAMP52 and CKAMP59 and their role in AMPA receptor function.

## 2. Structure of CKAMP Family Members

All CKAMP family members are type 1 transmembrane proteins with a signal peptide, an N-terminal cysteine-rich extracellular domain, a single transmembrane domain and a large intracellular C-terminal domain with a terminal PDZ type II motif (Figure 1A).

The extracellular cysteine-rich domains of CKAMP family members are highly conserved between species, suggesting that they are of importance for protein function. Disulfide bonds between these cysteines may stabilize a globular structure that is important for the interaction with or modulation of AMPA receptors. Indeed, CKAMP44 variants with individual cysteines mutated to alanine do not modulate AMPA receptor function. Binding to AMPA receptors, however, was not affected by the mutations. This strongly suggested that an interaction of an extracellular disulfide bond-stabilized structure of CKAMP44 with an extracellular structure of AMPA receptors is responsible for the modulation exerted by this auxiliary protein [16]. The cysteine-rich domains of the other CKAMP family members exhibit a particularly high degree of similarity with that of CKAMP44 (Figure 1) [12], suggesting that they serve similar functions. Based on the disulfide bonds of similar cysteine-rich domains in other proteins, Pei & Grishin predicted disulfide bond patterns for shisa proteins and CKAMP/shisa family members [17].

Not only the cysteine-rich region, but also the transmembrane region and a short stretch of 20 amino acids downstream of the transmembrane region show a high level of homology between CKAMP family members (Figure 1). Deletion of the 20 intracellular amino acids abolished the interaction of CKAMP44 with AMPA receptors, whereas a deletion of the entire intracellular C-terminus but these 20 amino acids did not affect interaction with or modulation of AMPA receptors [16]. These data suggest that this domain is responsible for the interaction with AMPA receptors. Two CKAMP44 mutants in which only 6 of the 20 amino acids were deleted (180–185 and 186–191) did also not interact with GluA1, suggesting that amino acids 180–191 are responsible for the interaction with AMPA receptors. It is of course also possible that these amino acids do not directly interact with AMPA receptors, but that their deletion affects tertiary structure or protein topology, which impairs the interaction with AMPA receptors. CKAMP44 and CKAMP52 can be found in AMPA receptor complexes that also contain TARPs [14,16], suggesting that CKAMPs and TARPs bind to distinct regions of the GluA subunits.

A third highly conserved region of 9 amino acids is present at the C-terminal end of all CKAMP proteins. The same PDZ type II binding motif, namely EVTV, is found at the very end of the C-terminal domain of all CKAMP family members. Domain deletion studies and yeast-two-hybrid screens showed that the EVTV motif serves as an interaction site for CKAMP family members with PDZ-domain containing proteins such as DLG1/SAP97, DLG2/PSD93, DLG3/SAP102, DLG4/PSD95, MAGI1, MAGI3, MPP5, GRIP1, PICK1, and Lin7b [13,16,18]. Interestingly, activation of PKC1 promoted the interaction of PICK1 with CKAMP44 and resulted in a phosphorylation of CKAMP44 [18]. The functional consequence of this interaction is still not understood at the moment.

There are two splice variants of CKAMP44, CKAMP52 and CKAMP59, whereas only one splice variant exists for CKAMP39 [5,12]. The short forms of CKAMP52 and CKAMP59 are more abundant in the brain than the long forms [12]. The alternatively spliced exons (CKAMP52: exon 3, CKAMP44 and CKAMP59: exon 4) encode for a short stretch of intracellular amino acids (16 in CKAMP44, 32 in CKAMP52, and 17 in CKAMP59) downstream of the AMPA receptor-interacting domain and upstream of the PDZ motif. The function of the amino acids that are encoded by the alternatively spliced exons is so far unknown.

We named CKAMP44 according to the predicted disulfide bond stabilized extracellular structure, the modulation of AMPA receptors and the predicted molecular weight (cysteine-knot AMPA receptor modulating protein, 44, kD) [5]. At the time when we identified the proteins of the CKAMP family their genes had not yet been recorded in the Refseq database. The genes were later named *shisa6* (encoding CKAMP52), *shisa7* (encoding CKAMP59), *shisa8* (encoding CKAMP39), and *shisa9* (encoding CKAMP44) based on similarities with other *shisa* genes. However, the genes encoding for CKAMP family members form a separate cluster from the shisa1–5 cluster on the phylogenetic tree (Figure 1) [12,17]. Shisa1–shisa5 do not contain the AMPA-interacting region of CKAMP family members [12,17] and were not identified as AMPA receptor interacting proteins in proteomic studies [2,3,4,5]. In addition, they also do not contain the functional important PDZ type II binding motif EVTV of the CKAMP family members [12,17]. These dissimilarities are consistent with the fact that shisa proteins serve different functions than CKAMP family members. Shisa1–shisa3 for instance reduce Wnt and FGF signaling by inhibiting the export of their receptors from the endoplasmic reticulum [19,20]. Interestingly, the cysteine-rich domain is conserved in *shisa* genes. This suggests that the role of the cystine-rich domains of shisa1–shisa5 resembles that of the CKAMP cystine-rich domain by mediating the modulation that they exert on the function of other proteins. In conclusion, there are structural similarities between shisa1–shisa5 and CKAMP family members, but shisa1–shisa5 lack several of the domains that are important for the function of CKAMP family members. Considering the differences in protein function (see below) it is obvious that CKAMP family members constitute a protein family distinct from the shisa protein family.

## 3. Expression of CKAMP Family Members

CKAMP family members are by and large brain-specific proteins [12,14,15] (see also BioGPS database). Within the brain, they display discrete and complementary regional expression profiles (Figure 2) [12]. The expression of CKAMP39 mRNA is restricted to the granule cell layers of cerebellum and olfactory bulb. CKAMP44 mRNA is found in many brain areas with a particularly high expression in the dentate gyrus and in the glomerular layer of the olfactory bulb. CKAMP52 mRNA is expressed in the principal cell layers of all hippocampal sub-regions, in the septum and in the Purkinje layer of the cerebellum. Finally, CKAMP59 mRNA is detected mainly in the cortex, striatum, principal cell layers of all hippocampal sub-regions, and granule cell layer of the olfactory bulb. mRNA analyses at different developmental stages show that CKAMP44 and CKAMP59 are expressed already prenatally. Postnatally, there is little change in expression levels, except for an upregulation of CKAMP39 and CKAMP52 in cerebellum and olfactory bulb and a modest downregulation of CKAMP59 in thalamus and brainstem [5,12]. AMPA receptors are expressed not only in neurons but also in astrocytes. They additionally express auxiliary subunits such as TARPs and, especially in cerebellar and cortical astrocytes, CKAMP44 (but not CKAMP52) [21].

Quantifications of the molecular abundance of interacting proteins in AMPA receptor complexes from different brain areas found expression profiles for CKAMP44 and CKAMP52 (CKAMP39 and CKAMP59 were not investigated in these studies) that are consistent with the mRNA expression analysis [22,23]. CKAMP52 protein is detectable mainly in the hippocampus and cerebellum, albeit at a rather low molecular abundance. CKAMP44 is found in many brain areas and at considerably higher levels than CKAMP52. Interestingly, CKAMP52-containing AMPA receptors do not contain CKAMP44 [14] despite the fact that both proteins are expressed in hippocampal neurons. CKAMP52 and CKAMP59 on the other hand are found in the same AMPA receptor complex [15].

The uniqueness of the expression profiles with comparably little overlap suggests that a given neuron type expresses at high levels only one or two CKAMP family members. This is an observation that has been made for other AMPA receptor interacting proteins as well [22]. Thus, each neuron type expresses a specific combination of GluA subunits and interacting proteins to obtain AMPAR receptor complexes with functions that are optimized for the synaptic computations of this particular neuron type.

## 4. Function of CKAMP Family Members

AMPA receptor interacting proteins serve different functions such as controlling receptor assembly, promoting export of assembled AMPA receptors from the endoplasmic reticulum and trafficking to the cell surface, influencing subcellular localization (e.g., synaptic and extrasynaptic localization) and finally modulating gating properties of AMPA receptors [6,7].

### 4.1. Assembly and Surface Trafficking of AMPA Receptors

The surface expression of AMPA receptors changes in neurons in which the expression of CKAMP family members is decreased or increased. AMPA receptor current amplitudes are reduced in dentate gyrus granule cells and lateral geniculate nucleus relay neurons of CKAMP44-deficient mice. Overexpression of CKAMP44 has the opposite effect [16,24]. How CKAMP44 promotes the surface expression of AMPA receptors is unclear. Several mechanisms are possible: AMPA receptors have to pass a stringent quality control before they can exit the endoplasmic reticulum (ER) [25]. AMPA receptor auxiliary subunits have been shown to promote ER exit by modulating AMPA receptor assembly, by influencing posttranslational modifications (e.g., glycosylation), by containing ER export signal motifs, by masking the ER retention signals of AMPA receptors, and by influencing the transport of AMPA receptors via the Golgi network to the cell surface [11,26,27,28,29,30,31,32,33]. In addition, AMPA receptor interacting proteins can serve as cargo proteins that promote the transport of AMPA receptors via the Golgi network to the cell surface [31]. CKAMP family members may influence the number of AMPA receptors on the cell surface of neurons by similar mechanisms. ER export depends on gating kinetics of AMPA receptors [33]. This suggests, that the quality control machinery preferentially allows gating-competent AMPARs to exit the ER [33]. It is thus reasonable to hypothesize that auxiliary subunits that modulate AMPA receptor gating should also influence ER exit by changing receptor conformation to one that is more or less preferred by the quality control machinery. Indeed, most auxiliary subunits that modulate gating also affect the trafficking of AMPA receptors to the cell surface [2,3,4,12,27,32] Consistent with this hypothesis is the observation that CKAMP44 mutants that do not influence AMPAR gating also do not promote the trafficking of receptors to the cell surface [16]. The influence of CKAMP family members on AMPA receptor gating shows GluA-subunit-specific differences (Table 1, see below). This may explain subunit-specific or cell type-specific differences in the influence of a given CKAMP family member on receptor trafficking [5,12,14,15,16].

Genetic deletion of CKAMP52 [14] and CKAMP59 [15] does not affect the abundance of synaptic AMPA receptor subunits or AMPA receptor-mediated current amplitudes in CA1 neurons. Somewhat contrasting effects were observed in HEK293 cells in which the co-expression of CKAMP52, CKAMP59 and also CKAMP39 decreases surface expression of AMPA receptors and AMPA receptor-mediated current amplitudes [12]. As discussed above, the influence of CKAMP family members shows AMPA receptor subunit-specific differences (Table 1). The influence of CKAMP52 and CKAMP59 on AMPA receptor trafficking in CA1 cells may therefore differ from the influence in HEK293 cells because of differences in the composition of AMPA receptors. In addition, native AMPA receptors interact also with other auxiliary subunits such as TARPs or cornichons. It has been shown that the influence of a given auxiliary subunit on trafficking or gating depends on the presence of other AMPA receptor interacting proteins [16,28,30]. Differences in the influence of a given CKAMP family member on receptor trafficking in heterologous cells such as HEK293 cells compared to the influence in neurons may of course also result from differences in the ER export machinery of the cell types. Thus, there is some uncertainty when predicting the role of an AMPA receptor auxiliary subunit in neurons based on results from experiments with heterologous cells such as *Xenopus laevis* oocytes or HEK293.

CKAMP family members may influence synaptic currents either by directly modulating gating of AMPA receptors (see below) or by affecting the composition of synaptic AMPA receptors. The fact that the influence of CKAMP family members on AMPA receptor trafficking depends on the GluA subunit suggests that CKAMP family members possibly affect the composition of synaptic AMPA. It seems likely that the different CKAMP family members bind at the same position in AMPA receptors. The interaction of one CKAMP family member may in this case be inhibited by another. In addition, binding to CKAMP family members may reduce the binding of AMPA receptors to other auxiliary subunits, similar to the reduction of TARP-AMPA receptor interaction by CNIH-2 [29]. Analysis of the AMPA receptor proteome showed that there are preferred co-assemblies of certain GluAs and auxiliary subunits. For example, CKAMP44 seems to preferentially co-assemble with GluA4, GSG1L and PSD95 [22]. However, currently there is little direct evidence for an effect of CKAMP family members on AMPA receptor composition.

### 4.2. Subcellular Localization of AMPA Receptors

CKAMP44, CKAMP52 and CKAMP59 are contained in synaptic AMPA receptors [5,14,15]. Moreover, deletion of CKAMP44 reduces the number of synaptic AMPA receptors [16]. The PDZ domain interacting motif EVTV is required for the CKAMP44-mediated synaptic localization of AMPA receptors. Thus, a CKAMP44 mutant in which EVTV was deleted did not augment synaptic AMPA receptor number, but still promoted the trafficking of AMPA receptors to the cell surface [16]. It is likely that the EVTV-mediated interaction of CKAMP44 with proteins such PSD95, SAP97, PSD93, and SAP102 stabilizes the synaptic localization of AMPA receptors. Synaptic AMPA receptor number is unaffected in CKAMP52^−/−^ and CKAMP59^−/−^ mice in contrast to the reduction in CKAMP44^−/−^ mice (Table 3) [14,15,16]. However, quantitation of diffusion showed that CKAMP52 reduces the lateral mobility of AMPA receptors at the cell surface. This influence also depends on the presence of the EVTV motif [14]. These findings are consistent with the hypothesis that CKAMP family members stabilize AMPA receptors at certain localizations in the membrane through interactions with intracellular PDZ domain-containing scaffolding proteins. Immunohistochemical experiments show strong CKAMP52 signal in hippocampal layers containing principle cell dendrites (e.g., stratum radiatum), but only a very weak signal in the cell body layers of dentate gyrus, CA1, CA2, and CA3 [14]. This indicates that CKAMP52 is part of dendritic but not somatic AMPA receptors. Consistently, the genetic deletion of CKAMP52 affected the function of dendritic (synaptic and extrasynaptic) but not the function of somatic AMPA receptors [14]. This is in contrast to CKAMP44 that influences synaptic and somatic currents [5,16,24].

### 4.3. Gating of AMPA Receptors

Experiments performed with heterologous cells show that all CKAMP family members influence AMPA receptor properties [12,14,15]. The influence is complex, depends on the AMPA receptor subunit and differs between CKAMP family members (Table 1). They affect AMPA receptor kinetics (rise time, deactivation, desensitization, recovery from desensitization rates), current amplitudes (peak, steady-state), rectification, apparent glutamate affinity, kainate efficacy and cyclothiazide affinity (Table 1). The influence of CKAMP44, CKAMP52, and CKAMP59 in neurons were by and large consistent with the influence in HEK293 cells or *Xenopus laevis* oocytes (Table 1, Table 2 and Table 3). There are subtle differences in their functions in heterologous cells and neurons. For example, CKAMP44 does not influence deactivation rate of GluA1 in HEK293 cells, but genetic deletion of CKAMP44 increases the deactivation rate in dentate gyrus granule cells. The influence of CKAMP family members also varies between different neuron types. For example, deactivation rate is altered in dentate gyrus granule cells but not in lateral geniculate nucleus relay neurons of CKAMP44^−/−^ mice [24]. As discussed above, the influence of CKAMP family members shows subunit specific differences (Table 1). Moreover, the composition of AMPA receptors varies in different cell types. It is therefore not too surprising that there is also a neuron-specificity in the influence of CKAMP family members on AMPA receptor-mediated currents.

Table 1 shows that the effects of CKAMP family members can also differ in the same heterologous cell type (HEK293 cells). This may result from different co-expressed GluA subunits (flip vs. flop, mouse vs. rat) or from different CKAMP isoforms (long vs. short). For example, Klaassen et al. found that the exon 3-containing CKAMP52 isoform decreases deactivation rate of GluA1 homomeric AMPA receptors [14], whereas we observed no influence of the exon 3-lacking CKAMP52 on deactivation [12]. Thus, it is possible that the intracellular amino-acids that are coded by exon 3 affect the function of CKAMP52.

### 4.4. Synaptic AMPA Receptor-Mediated Currents

Studies with knockout mice and virus-mediated protein overexpression showed that CKAMP family members influence synaptic AMPA receptor-mediated currents. Genetic deletion of CKAMP44 decreases current amplitude [16,24], suggesting that this auxiliary subunit increases the number of synaptic AMPA receptors (see above). In contrast, synaptic current amplitude is not affected in CKAMP52^−/−^ and CKAMP59^−/−^ mice (Table 3). Interestingly, genetic deletion of CKAMP52 and CKAMP59 decreased decay times of mEPSCs [14,15]. Thus, CKAMP52 and CKAMP59 influence gating but not the number of synaptic AMPA receptors. The mEPSC decay rate is unaltered in dentate gyrus granule cells of CKAMP44^−/−^ mice. At first glance, this may be unexpected because the deactivation rate of extrasynaptic AMPA receptor-mediated currents is increased in CKAMP44^−/−^ mice [16]. However, granule cells also express TARPγ-8 which, similarly to CKAMP44, decreases the deactivation rate of extrasynaptic, but not decays, of synaptic AMPA receptor-mediated currents [16]. What could be the reason that the genetic deletion of a subunit affects the deactivation of extrasynaptic AMPA receptor-mediated currents but not the decay of synaptic currents? The most likely explanation is that the genetic deletion of CKAMP44 or TARPγ-8 reduces synaptic AMPA receptor number, and that the remaining AMPA receptors still interact with the other auxiliary subunits that exert a similar effect on AMPA receptor deactivation.

### 4.5. Synaptic Short-Term Plasticity

CKAMP44 and CKAMP52 modulate synaptic short-term plasticity (Table 3). Thus, the genetic deletion of CKAMP44 increases short-term facilitation (medial perforant path) (Figure 3) and reduces short-term depression (lateral perforant path) in dentate gyrus granule cells [16]. Similarly, short-term depression is reduced in retinogeniculate synapses of lateral geniculate nucleus relay neurons from CKAMP44^−/−^ mice [24]. In contrast, deletion of CKAMP52 decreases short-term facilitation in synapses of CA1 neurons [14]. Unaltered short-term plasticity of NMDA receptor-mediated currents showed that the observed changes did not result from an influence of CKAMP44 or CKAMP52 on presynaptic function (release probability). The mechanisms of how CKAMP44 and CKAMP52 influence short-term plasticity are very different. Thus, CKAMP44 affects short-term plasticity by increasing desensitization and by decreasing the recovery from desensitization rate [5,16,24]. CKAMP44 therefore increases the likelihood that AMPA receptors are in the desensitized state when a synapse is repetitively activated. Consistent with this interpretation is the observation that CKAMP44 exerts no influence on short-term plasticity when AMPA receptor desensitization is blocked with cyclothiazide [16]. We investigated the relevance of the influence of CKAMP44 on synaptic short-term plasticity in more detail in the lateral geniculate nucleus. Current clamp experiments indicated that CKAMP44 reduces firing probability of relay neurons when they receive trains of inputs from the optic tract. This reduction was mediated by the influence of CKAMP44 on synaptic short-term plasticity. Importantly, in vivo recording of relay neuron activity of head-fixed mice in response to visual stimuli confirmed that CKAMP44 reduces firing probability [24]. This shows that auxiliary subunits influence the integration of excitatory inputs not only by affecting synaptic strength, but also by modulating synaptic short-term plasticity.

The influence of CKAMP52 on the recovery from desensitization rate depends on the AMPA receptor subunit (Table 1). CKAMP52 decreases the recovery from desensitization of GluA1/2 heteromeric receptors [14], which is the dominant receptor subunit combination in synapses of CA1 neurons [35]. This suggests that the modulation of recovery from desensitization is not the relevant factor for the influence of CKAMP52 on synaptic short-term plasticity in CA1 neurons (one would expect in this case the opposite influence on short-term plasticity). Instead, CKAMP52 increases short-term facilitation most likely by decreasing the rate of deactivation and desensitization and by increasing steady-state current amplitudes of AMPARs [14]. A change in the deactivation or desensitization rate would be relevant only if synapses are activated with sufficiently high frequencies. The slowing or saturation of glutamate reuptake during repetitive high frequency activity may also increase extracellular glutamate concentrations to levels where a CKAMP52-mediated increase in steady-state current amplitude could affect short-term plasticity. Indeed, CKAMP52 influences short-term plasticity only if CA1 synapses are activated with a train of stimuli at high frequency (50 Hz, but not 2 or 20 Hz) [14]. This strengthens the hypothesis that CKAMP52 reduces short-term depression by stabilization of the open state of the receptors.

Diffusion of AMPA receptors at the cell surface reduces the impact of desensitization on short-term plasticity because desensitized synaptic receptors are quickly replaced by non-desensitized receptors [36]. Proteins such as TARPs and CKAMP family members that reduce diffusion of AMPA receptors [14,37] may therefore be expected to increase short-term depression. It is thus possible that CKAMP44 increases short-term depression (or reduces facilitation) not only by slowing the recovery from desensitization but also by stabilizing synaptic AMPA receptors. CKAMP52, on the other hand, may increase facilitation by stabilizing synaptic AMPA receptors that have a lower deactivation rate and that are less sensitive to desensitization than AMPA receptors that do not bind to CKAMP52 [14]. The influence of TARPs on short-term depression is reduced by their quick dissociation from desensitized AMPA receptors, which therefore diffuse out of synapses such that they can be replaced by non-desensitized receptors [38]. It is unclear if a similar mechanism modulates the influence of CKAMP52 on short-term plasticity.

### 4.6. Long-Term Plasticity

Auxiliary subunits such as TARPs, cornichons, and GSG1L play a prominent role in synaptic long-term plasticity [30,39,40,41]. The mechanisms of how TARPs influence synaptic plasticity have been investigated in more detail in several studies, which showed that phosphorylation-mediated changes in the influence of TARPs on AMPA receptor diffusion and trapping of AMPA receptors in synapses play a role [40,42,43]. Considering that CKAMP family members also modulate diffusion and stabilize AMPA receptors in synapses, and additionally influence their trafficking to the cell surface [5,12,14,16], it seems quite likely that they also play a role in synaptic long-term plasticity. Long-term potentiation (LTP) is indeed absent in CA1 neurons of CKAMP59^−/−^ mice, however the exact mechanism is still not known [15]. CKAMP44, on the other hand, is not required for LTP, at least in dentate gyrus granule cells. LTP depends in these cells on the presence of TARPγ-8 [16].

### 4.7. Behavior

Relatively little is known about the role of CKAMP family members in behavior. Their influence on AMPA receptors, synaptic function, long term plasticity, and integration of excitatory inputs suggests that behavior is affected in knockout mice. Indeed, contextual fear memory is severely impaired in CKAMP59^−/−^ mice. This is in line with the reduction of LTP in CA1 neurons of CKAMP59^−/−^ mice. Auditory fear memory is not affected by the deletion of CKAMP59, indicating that the protein is of importance for hippocampus-dependent, but not amygdala-dependent learning despite the fact that CKAMP59 is expressed also in this brain area [15]. Other behavioral analyses (base line locomotor activity, shock perception, open field, elevated plus maze, dark-light box) showed no abnormalities in CKAMP59^−/−^ mice [15]. So far, there is no published study on the behavior in knockout mice of the other CKAMP family members.

### 4.8. Other Functions

A recent study suggests that CKAMP52 inhibits Wnt signaling in the testis [44]. This is reminiscent of the function of shisa1–3 that reduce Wnt and FGF signaling by inhibiting the export of their receptors from the endoplasmic reticulum [19,20]. A role of CKAMP52 in the testis is, however, somewhat at odds with the absence of CKAMP52 mRNA expression in the testis [14]. Further studies are needed to show if the function of CKAMP family members is primarily to influence AMPA receptors, or if they have in addition functions similar to that of shisa1–5.

## 5. Conclusions

CKAMP39, CKAMP44, CKAMP52, and CKAMP59 constitute a recently identified family of AMPA receptor interacting proteins that exert diverse influences on AMPA receptor expression and function. The studies of CKAMP family members that were published so far showed that CKAMP44, CKAMP52 and CKAMP59 are bona fide AMPA receptor auxiliary subunits. The findings from experiments with heterologous cells further suggest that CKAMP39 is an AMPA receptor auxiliary subunit. However, it remains to be shown whether CKAMP39 has a function in the brain. More research on the function of all CKAMP family members in the brain is warranted. Several important questions remain open: Do CKAMP family members alter GluA subunit composition by affecting trafficking in a subunit-specific manner? What is the mechanism by which CKAMP59 (and perhaps other CKAMP family members) modulates LTP? The influence of a given auxiliary subunit depends on the GluA subunit composition and is modulated by other auxiliary subunits (see e.g., [16]). It is therefore quite likely that the influence of an auxiliary subunit as observed in heterologous cells, in which usually only a certain combination of GluA and auxiliary subunits is expressed, differs from the influence in a particular neuron type that expresses AMPA receptors that are built of a complex combination of GluA and auxiliary subunits. Hence, for a better understanding of the role of an auxiliary subunit, it is mandatory to manipulate the expression of this protein in the brain, by using for example knockout mice, and to investigate the effect of this manipulation on the function of native AMPA receptors.

## Figures and Tables

**Figure 1 ijms-20-01460-f001:**
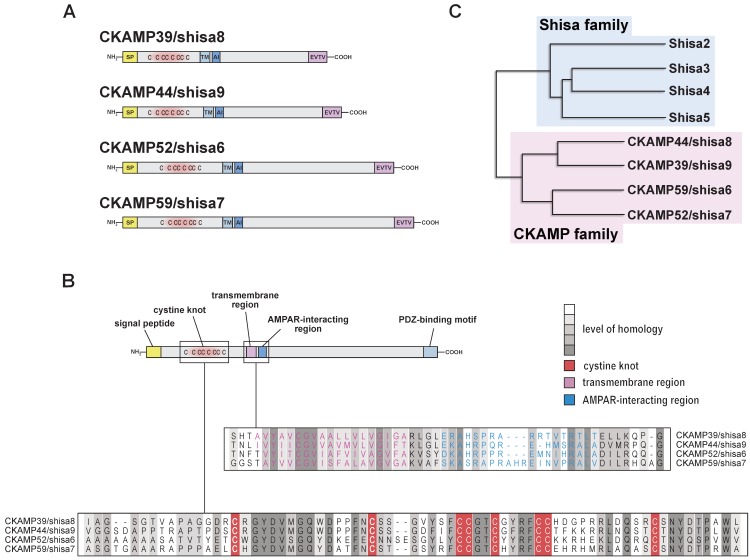
(**A**) Schematic drawing of CKAMP family members with signal peptide (SP), cysteines (C) of the cystine knot, transmembrane domain (TM), AMPA receptor interacting region (AI), and PDZ type II binding motif (EVTV). (**B**) Protein sequence alignment of the cystine-rich domain and the transmembrane region of mouse CKAMP39, CKAMP44, CKAMP52 and CKAMP59. Amino acids marked in grey are similar or identical among CKAMP family members. Intensity of the grey color indicates the degree of similarity. Red, violet, and blue letters mark amino-acids belonging to the cystine-rich domain, transmembrane and AMPA receptor-interacting region. (**C**) Phylogenetic tree of shisa proteins and CKAMP/shisa family members, based on their protein sequence (average distance tree). (Adapted from Figure 1, Farrow et al., Auxiliary subunits of the CKAMP family differentially modulate AMPA receptor properties. *elife* 2015, *4*, e09693, doi:10.7554/eLife.09693 [12]).

**Figure 2 ijms-20-01460-f002:**
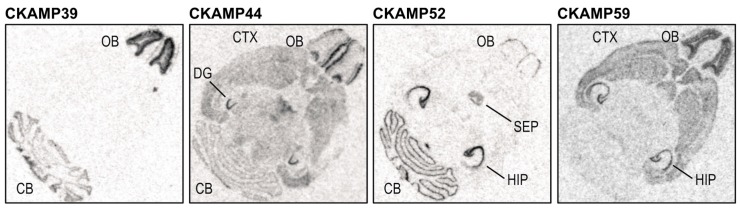
In situ hybridization of horizontal brain sections from adult mice showing the distinct expression profiles of the four CKAMP family members. OB = olfactory bulb; CB = cerebellum; CTX = cortex; DG = dentate gyrus; SEP = septum; HIP = hippocampus. (Adapted from Figure 2, Farrow et al., Auxiliary subunits of the CKAMP family differentially modulate AMPA receptor properties. *elife* 2015, *4*, e09693, doi:10.7554/eLife.09693 [12]).

**Figure 3 ijms-20-01460-f003:**
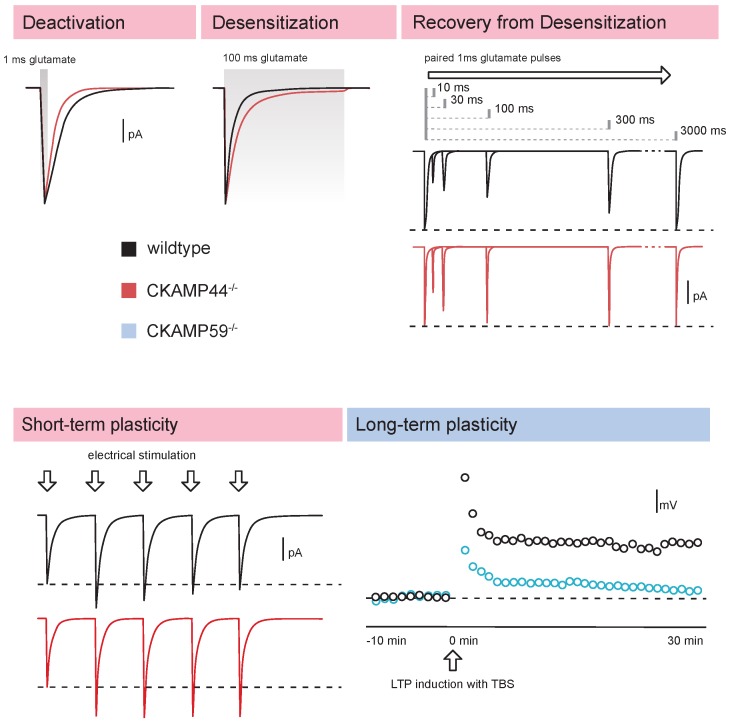
Schematic drawings of currents illustrating influences of CKAMP family members on AMPA receptor gating properties, synaptic short-term and long-term plasticity. Upper panels: Genetic deletion of CKAMP44 increases deactivation rate, decreases desensitization rate and increases the rate of recovery from desensitization of AMPA receptor-mediated currents. Extrasynaptic AMPA receptors were activated by ultrafast application of 1 ms (deactivation, recovery from desensitization) or 100 ms (desensitization) glutamate-pulses onto outside-out patches of dentate gyrus granule cells. Adapted from [16] with permission. Lower panels: CKAMP44 modulates short-term plasticity as evidenced by the increased short-term facilitation in perforant path synapses of CKAMP44^−/−^ mice. Adapted from [16] with permission. Long-term potentiation is absent in CA1 neurons of CKAMP59^−/−^ mice showing the involvement of this CKAMP family member in synaptic plasticity. LTP was induced with a theta burst stimulation (TBS) protocol. (Adapted from Figure 6, Schmitz et al., The AMPA receptor-associated protein Shisa7 regulates hippocampal synaptic function and contextual memory. *elife* 2017, *6*, e24192, doi:10.7554/eLife.24192) [15].

**Table 1 ijms-20-01460-t001:** Modulation of AMPA receptor expression, gating in HEK293 cells, oocytes and cultured neurons.

	CKAMP39			CKAMP44			CKAMP52			CKAMP59		
	A1	A2	A1/2	A1	A2	A1/2	A1	A2	A1/2	A1	A2	A1/2
Surface expression	↓ [12]	↓ [12]	N/A	↑ [16]	N/A	N/A	- [12]	↓ [12]	N/A	↓ [12]	↓ [12]	N/A
Current amplitude	↓ [12]	↓ [12]	N/A	↓ [34]	N/A	N/A	- [12]	↓ [12]	N/A	- [12]	↓ [12]	N/A
Rise time	- [34]	- [34]	N/A	- [34]	N/A	N/A	- [14]	- [14]	- [14]	- [15]	N/A	N/A
Deactivation rate	- [12]	-/↓ [12]	N/A	- [34]	N/A	N/A	-/↓ [12,14]	↓ [12,14]	↓ [14]	- [12,15]	- [12]	N/A
Desensitization rate	- [12]	↑ [12]	N/A	↑ [16]	N/A	N/A	-/↓ [12,14]	-/↑ [12]	↓ [14]	-/↑ [12,15]	- [12]	N/A
Steady-state current amplitude	- [12]	↓ [12]	N/A	↓ [16]	N/A	N/A	↑ [12,14]	↑ [12]	↑ [14]	-/↓ [12]	- [12]	N/A
Recovery from Desensitization rate	↓ [12]	↓ [12]	N/A	↓ [34]	↓ [16]	N/A	-/↓ [12]	-/↑ [12]	↓ [14]	-/↓ [12,15]	- [12]	N/A
Rectification	N/A	N/A	N/A	N/A	N/A	N/A	- [14]	N/A	- [14]	- [15]	N/A	N/A
Apparent glut affinity	↑ [12]	↑ [12]	N/A	↑ [5,12]	↑ [12]	N/A	↑ [12]	↑ [12]	N/A	N/A	N/A	N/A
Apparent CTZ affinity	N/A	N/A	N/A	↓ [5]	N/A	N/A	N/A	N/A	N/A	N/A	N/A	N/A
I_KA_/I_Glut_	N/A	N/A	N/A	N/A	N/A	↓ [21] *	N/A	N/A	N/A	N/A	N/A	N/A

A1 = GluA1, A2 = GluA2, A1/2 = GluA1/2, N/A = not available, I = current, KA = kainate, Glut = glutamate, CTZ = cyclothiazide. * tested with GluA1/4 heteromers.

**Table 2 ijms-20-01460-t002:** Influence of CKAMP family members on extrasynaptic (somatic) AMPA receptor-mediated currents.

	CK39	CK44	CK52	CK59
Extrasynaptic (somatic) AMPA receptor-mediated current amplitudes	N/A	↑ [16,24]	NA	N/A
Rise time	N/A	- [16,24]	- [14]	N/A
Deactivation rate	N/A	↓ * [5,16]	- [14]	N/A
Desensitization rate	N/A	↑ * [5,16]	NA/	N/A
Steady-state current amplitude	N/A	↓ [5,16,24]	N/A	N/A
Recovery from desensitization rate	N/A	↓ [5,16,24]	N/A	N/A
Rectification	N/A	↑ [16]	N/A	N/A
Conductance	N/A	↑ [16]	N/A	N/A
Apparent glutamate affinity	N/A	↑ [5,16]	N/A	N/A

N/A = not available. Currents were evoked by fast application of glutamate onto outside out patches or nucleated patches. Effects are deduced from the changes observed in CKAMP44^−/−^, CKAMP52^−/−^, and CKAMP59^−/−^ mice. [5]—Outside out (amplitude, rise time, deactivation) and nucleated patches (desensitization, steady state current, recovery from desensitization) of CA1 cells; [16]—Outside out patches of dentate gyrus granule cell; [24]—Nucleated patches of lateral geniculate nucleus relay neurons; [14]—Nucleated patches of CA1 cells. * Deactivation and desensitization rates were not affected by CKAMP44 in relay neurons of the lateral geniculate nucleus.

**Table 3 ijms-20-01460-t003:** Modulation of AMPA receptor-mediated currents by CKAMP family members.

	CK39	CK44	CK52	CK59
mEPSC amplitudes	N/A	↑ [16]	- [14]	- [15]
mEPSC frequency	N/A	↑ [16]	- [14]	- [15]
mEPSC rise time	N/A	- [16]	↑ [14]	- [15]
mEPSC decay time	N/A	- [16]	↑ [14]	↑ [15]
AMPA/NMDA ratio	N/A	↑ [16,24]	N/A	- [15]
Short-term plasticity (↑ = more facilitation)	N/A	↓ [5,16,24]	↑ [14]	- [15]
Long-term potentiation	N/A	- [16]	N/A	↑ [15]

N/A = not available. Effects are deduced from the changes observed in CKAMP44^−/−^, CKAMP52^−/−^, and CKAMP59^−/−^ mice. [5]—CA1 cells and dentate gyrus granule cell; [16]—dentate gyrus granule cell; [24]—lateral geniculate nucleus relay neurons; [14]—CA1 cells; [15]—CA1 cells.

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
