# Peer review of "AMPA Receptor Auxiliary Proteins of the CKAMP Family"

_ijms, 2019, doi:10.3390/ijms20061460_

Round 1

Reviewer 1 Report

Dr. von Engelhardt reports well about recent studies on the various roles of CKAMP family members in AMPAR function. This review will be of interest to the synapse biology field and a broad readership in molecular sciences. I have the following scientific and editorial comments that would enrich the current manuscript:

1. The AMPAR complex has been known to have many auxiliary protein families that affect surface trafficking, synaptic localization, and channel properties besides CKAMPs (e.g., TARPs, CNIHs, and GSG1L). Although it was commented in lines 74-75 and 380-382 like "The influence of a given auxiliary subunit depends on the GluA subunit composition and is modulated by other auxiliary subunits", additional discussion how CKAMP family proteins may change the AMPAR complex stoichiometry or the composition of other auxiliary protein families would make the manuscript much more fruitful.

2. In Figure 1C, indicating Shisa protein names next to the CKAMPs would be helpful for readers (for example, 'CKAMP44/Shisa9').

3. In Figure 3, I'd suggest specifying the LTP induction protocol (e.g., TBS).

4. CKAMP family proteins were mis-labeled in the followings:
Lines 35-36; They should be "CKAMP52 and CKAMP59".
Line 131; It should mean "CKAMP44 and CKAMP52".

5. There were a few typos:
Line 383; "auxiliary"
Line 388; "native" or "naive"

Author Response

Reviewer 1

Dr. von Engelhardt reports well about recent studies on the various roles of CKAMP family members in AMPAR function. This review will be of interest to the synapse biology field and a broad readership in molecular sciences. I have the following scientific and editorial comments that would enrich the current manuscript:

1. The AMPAR complex has been known to have many auxiliary protein families that affect surface trafficking, synaptic localization, and channel properties besides CKAMPs (e.g., TARPs, CNIHs, and GSG1L). Although it was commented in lines 74-75 and 380-382 like "The influence of a given auxiliary subunit depends on the GluA subunit composition and is modulated by other auxiliary subunits", additional discussion how CKAMP family proteins may change the AMPAR complex stoichiometry or the composition of other auxiliary protein families would make the manuscript much more fruitful.

I added a paragraph about how CKAMP family proteins may influence AMPAR complex composition (line 251-262).

2. In Figure 1C, indicating Shisa protein names next to the CKAMPs would be helpful for readers (for example, 'CKAMP44/Shisa9').

I followed the reviewer’s suggestion

3. In Figure 3, I'd suggest specifying the LTP induction protocol (e.g., TBS).

I specified in the Figure and legend that a TBS protocol was used.

4. CKAMP family proteins were mis-labeled in the followings:
Lines 35-36; They should be "CKAMP52 and CKAMP59".
Line 131; It should mean "CKAMP44 and CKAMP52".

I have corrected the mistakes.

5. There were a few typos:
Line 383; "auxiliary"
Line 388; "native" or "naive"

I have corrected the mistakes.

Reviewer 2 Report

This paper gives an excellent overview of the relatively recently studied SHISA/CKAMP family of proteins. The review is concise and provides adequate insight and enough depth into the various aspects of AMPAR function that these auxiliary proteins are involved in.

There are however various key aspects that would need revision before publication.

Major revisions:

1. The author states that the extracellular domain of these proteins is highly conserved between species and that “similar cysteine-rich domains are found in proteins such as growth factors, toxins, and interleukins”. Unfortunately there is little evidence for this. There is no exact spacing of the CKAMP/SHISA cysteines found in other proteins, such as toxins, to my knowledge. For instance, the contoxins mentioned have a very tightly packed Cysteine structure made of only three pairs of cysteines with a completely different spacing.  I invite the author to shows where conservation is located by adding an alignment of the mentioned proteins to convince this reviewer. Otherwise these statements need to be removed.

2. The author mentions that the CKAMP proteins contain a ‘cystine-knot’ by forming disulfide bonds between the 8 highly conserved extracellular cysteines.. This incorrect as a cystine knot is defined as a 3-disulphide bond structure of well-known three dimensional structure, whereas the proteins at hand contain 4 pairs of cysteines and without any data on their disulphide bridging pattern. A cysteine knot even with 3 cysteine pairs cannot be argued as the links between the cysteines should be known for this. To get educated on the cystine knot I recommend reading “Bioactive cystine knot proteins” by Norelle Daly and David Craik, Current Opinion in Chemical Biology, 15(3), 362–368 (2011).

3. The author does not comply with the nomenclature for gene and protein names. I checked, ENSEMBLE, NCBI, and UNIPROT databases and none of these have the CKAMPs listed as Gene/protein ID. They are sometimes mentioned as aliases. Given that the author has been publishing with the alias names I could understand mentioning the alias after a slash (SHISAx/CKAMPx). However, it cannot be accepted that a leading review on this protein family does not mention the official names for these genes. Please correct this omission throughout the manuscript.

4. In line with point 3. The author states “We named the CKAMP family members according to the functionally important cysteine-knot [for which I disagree, see point 2], their interaction with and modulation of AMPA receptors [correct] and their predicted molecular weight [I disagree]. Regarding the latter: unfortunately the CKAMP numbers do not at all correspond to their molecular weights. For instance, CKAMP52 is 66 KDa, CKAMP39 is 42.4 kDa, etc. The author should delete this statement (or alternatively assign the proper molecular weights to the CKAMPs).

5. Figure 1C makes mention of two parts of the SHISA family.  In accordance with point 3: this should be changed into 1 single title with the ‘SHISA/CKAMP’ family and all family members should get their adequate names:  ‘SHISAx/CKAMPx’.

6. Lines 91-110: The authors make the point that the SHISA subgroup is different from the AMPAR interacting CKAMP family members. This maybe true for the AMPAR binding (although this has formally not been tested), but the SHISA family nomenclature (see above) cannot be ignored by the author. This means that lines 91 to 110 need to be revised in order to get a proper description of the family. (i.e., naming SHISA members and SHISA/CKAMP members, instead of SHISA versus CKAMP members). Also, to my knowledge it is currently not clear whether the SHISA8/CKAMP39 member of the family has a role in modulating AMPARs in the brain. This claim should therefore await further evidence.

7. The authors state that CKAMP44 mRNA is found in many brain areas with a particularly high expression in dentate gyrus granule cells and in periglomerular cells of the olfactory bulb. However, recent single cell sequencing data does not show prove of this. In fact, it shows that SHISA9/CKAMP44 in the mouse brain is expressed most abundantly in inhibitory neurons. The author should make mention of this as it currently is misleading the reader. (For the author: please check http://linnarssonlab.org/cortex/ and type in: Shisa9 based on PMID:25700174)

8. The author states that: “The uniqueness of the expression profiles with comparably little overlap suggests that a given neuron type expresses only one or two CKAMP family members. This is an observation that has been made for other AMPA receptor interacting proteins as well [25]”. There is to my knowledge no convincing data that brings evidence for the separate expression or the colocalization of the SHISA/CKAMP proteins.  The author should be clear about this.

Minor comments

Figure 2. It is a pity that the author is using this low resolution in situ hybridization data. This does not do justice to the information from better types of expression data in Allen Brain Atlas and the recent single cell sequencing data for mouse brain. The latter data also provide a much better insight into expression of the SHISA/CKAMP family than the in situ hybridisations. If the in situ hybridization of the author is kept, than please use better resolution (as the author did previously) and indicate the structures in the image (as the author did previously).

Line 61: “Indeed, CKAMP44 variants with individual cysteines mutated to alanine could not modulate AMPA receptor function.”  I have the impression the author wants to say the reverse “…..individual cysteines mutated to alanine could modulate AMPA receptor function.

Line 73: “These data showed that this domain is responsible for the interaction with AMPA receptors.” Deletion of 20 amino acids is quite substantial and may well affect tertiary structure or protein topology by which interaction with the AMPAR is lost.  This alternative should be mentioned.

Line 136: “Interestingly, CKAMP52-containing AMPA receptors do not contain CKAMP44 despite the fact that both proteins are expressed in hippocampal neurons.” I think the author needs to report that SHISA9/CKAMP44 is most abundantly expressed in inhibitory neurons in the brain, whereas SHISA6/CKAMP52 is most abundantly expressed in excitatory cells. Again, please refer to novel insight from single cell RNA sequencing (see above, PMID:25700174).

Figure 3. There is something wrong with the spacing of the lettering in the figure. They space very unequally. Please repair.

Line 330: “In  contrast, the reduction in receptor diffusion cannot be the relevant mechanism by which CKAMP52 increases facilitation.” This is true, but Klaassen et al made the case that the stabilized receptors are less sensitive to desensitization, which would decrease the need for lateral diffusion. It would be good to sketch theis entire line of reasoning here.

Line 383: ‘auxiliary’ is misspelled.

Author Response

I would like to thank the Reviewer for his/her critical comments and thoughtful suggestions, which lead to an essential improvement of the manuscript. I hope that the new version will find the reviewers’ approval. 

Reviewer 2

This paper gives an excellent overview of the relatively recently studied SHISA/CKAMP family of proteins. The review is concise and provides adequate insight and enough depth into the various aspects of AMPAR function that these auxiliary proteins are involved in. 

There are however various key aspects that would need revision before publication. 

Major revisions:

1. The author states that the extracellular domain of these proteins is highly conserved between species and that “similar cysteine-rich domains are found in proteins such as growth factors, toxins, and interleukins”. Unfortunately there is little evidence for this. There is no exact spacing of the CKAMP/SHISA cysteines found in other proteins, such as toxins, to my knowledge. For instance, the contoxins mentioned have a very tightly packed Cysteine structure made of only three pairs of cysteines with a completely different spacing.  I invite the author to shows where conservation is located by adding an alignment of the mentioned proteins to convince this reviewer. Otherwise these statements need to be removed.

I removed the statement.

2. The author mentions that the CKAMP proteins contain a ‘cystine-knot’ by forming disulfide bonds between the 8 highly conserved extracellular cysteines.. This incorrect as a cystine knot is defined as a 3-disulphide bond structure of well-known three dimensional structure, whereas the proteins at hand contain 4 pairs of cysteines and without any data on their disulphide bridging pattern. A cysteine knot even with 3 cysteine pairs cannot be argued as the links between the cysteines should be known for this. To get educated on the cystine knot I recommend reading “Bioactive cystine knot proteins” by Norelle Daly and David Craik, Current Opinion in Chemical Biology, 15(3), 362–368 (2011).

It is true that it has not been shown that the cysteines form disulfide bonds and I rephrased that statement accordingly. However, the regular numbers of highly conserved cysteines suggests that they form disulfide bonds. The fact that CKAMP44 mutants in which individual cysteines are mutated into alanines are non-functional (Khodosevich et al., 2014) is consistent with the hypothesis that they form disulfide bonds. Based on the disulfide bonds of similar cysteine-rich domains, Pei & Grishin predicted disulfide bond patterns for shisa proteins (Pei & Grishin; Cell Sign 2012). 

By the way, there are cystine-knot containing proteins that stabilize a cystine-knot by forming disulfide-bonds in addition to the 3 disulfide bonds of the cystine-knot (see e.g. Darling et al., Cystine Knot Mutations Affect the Folding of the Glycoprotein Hormone-Subunit, JBC 2000). More than 6 cysteines therefore do not speak against the formation of a cystine-knot. However, it is true that we did not show that shisa/CKAMP proteins form a cystine-knot. I therefore deleted this statement.

3. The author does not comply with the nomenclature for gene and protein names. I checked, ENSEMBLE, NCBI, and UNIPROT databases and none of these have the CKAMPs listed as Gene/protein ID. They are sometimes mentioned as aliases. Given that the author has been publishing with the alias names I could understand mentioning the alias after a slash (SHISAx/CKAMPx). However, it cannot be accepted that a leading review on this protein family does not mention the official names for these genes. Please correct this omission throughout the manuscript.

Consistent with ENSEMBLE, NCBI, and UNIPROT databases, I had used shisa6, shisa7, shisa8, and shisa9 (and not CKAMP39, CKAMP44, CKAMP52 and CKAMP59) for the gene names. I also mentioned that there are two synonyms for the protein names. In the new version of the manuscript, I mention the shisa names also at a more prominent position of the review, i.e. already in the introduction. However, I think that using shisaX/CKAMPX throughout the review will reduce readability. I decided to use CKAMP39, CKAMP44, CKAMP52, and CKAMP59 as protein names throughout the review because these were the names that were used in the first publications of the proteins (von Engelhardt et al., Science 2010; Schwenk et al., Neuron 2014; Khodosevich et al., Neuron 2014). The genes encoding for the CKAMPs were named shisas after the proteins were named CKAMPs. Another reason for using CKAMP is that this name contains more information about the proteins (even when the predicted disulfide-bond structure is not a classical cystine-knot) than shisa (used due to some resemblance of the shisa1 expression with the form of a sculpture, common to southern Japan, with a large head similar to the Egyptian sphinx). For similar reasons, Cacngs (calcium channel, voltage-dependent, gamma subunits) are usually mentioned by their aliases (i.e. TARPs), and the alternative names SAP97, PSD93, SAP102, and PSD95 are mostly used for the DLG protein family. In addition, common alternative protein names are frequently used in reviews for a wider audience (e.g. GluA1-4 instead of gria1-4). However, to reduce possible confusions, I explained why two alternative names (i.e. shisas and CKAMPs) are currently used in the literature (line 133-152). 

4. In line with point 3. The author states “We named the CKAMP family members according to the functionally important cysteine-knot [for which I disagree, see point 2], their interaction with and modulation of AMPA receptors [correct] and their predicted molecular weight [I disagree]. Regarding the latter: unfortunately the CKAMP numbers do not at all correspond to their molecular weights. For instance, CKAMP52 is 66 KDa, CKAMP39 is 42.4 kDa, etc. The author should delete this statement (or alternatively assign the proper molecular weights to the CKAMPs).

I rephrased the statement accordingly.

A short explanatory note: The molecular weights especially of shisa8/CKAMP39 and shisa6/CKAMP52 were indeed and unfortunately miscalculated. CKAMP52/shisa6 was first published in Schwenk et al., Neuron 2014 and I do not know why they selected CKAMP52 as a name. We stuck to that name in the paper with the first description of the whole family (Farrow et al., eLIFE 2015). MWs of shisa7 and shisa8 were calculated by a Postdoc in the lab of Hannah Monyer who also found them in a database search. I asked her about the (mis)calculation of the MWs, but she did not recall details. If I remember correctly, the name CKAMP44 for shisa9 was proposed by Peter Seeburg and we discussed during that time if we should simply name it CKAMP or CKAMP-1, but decided to add 44, as it was not clear if there exist additional family members. In retrospect, simple numbering (CKAMP1-4) may have been the better choice (especially since posttranslational modifications change the MWs).

5. Figure 1C makes mention of two parts of the SHISA family.  In accordance with point 3: this should be changed into 1 single title with the ‘SHISA/CKAMP’ family and all family members should get their adequate names:  ‘SHISAx/CKAMPx’.

I updated Figure 1 and the legend of figure 1.

6. Lines 91-110: The authors make the point that the SHISA subgroup is different from the AMPAR interacting CKAMP family members. This maybe true for the AMPAR binding (although this has formally not been tested), but the SHISA family nomenclature (see above) cannot be ignored by the author. This means that lines 91 to 110 need to be revised in order to get a proper description of the family. (i.e., naming SHISA members and SHISA/CKAMP members, instead of SHISA versus CKAMP members). Also, to my knowledge it is currently not clear whether the SHISA8/CKAMP39 member of the family has a role in modulating AMPARs in the brain. This claim should therefore await further evidence.

See above as to the nomenclature. A added the information that for shisa8/CKAMP39 it has not been shown that it modulates AMPA receptor function in the brain (see line 459-461).

7. The authors state that CKAMP44 mRNA is found in many brain areas with a particularly high expression in dentate gyrus granule cells and in periglomerular cells of the olfactory bulb. However, recent single cell sequencing data does not show prove of this. In fact, it shows that SHISA9/CKAMP44 in the mouse brain is expressed most abundantly in inhibitory neurons. The author should make mention of this as it currently is misleading the reader. (For the author: please check http://linnarssonlab.org/cortex/ and type in: Shisa9 based on PMID:25700174)

I am aware of the study from the lab of Sten Linnarsson, but I have some doubts about the results for shisa9. As an example: according to their results shisa9 expression is double as high in CA1 neurons than in DG granule cells. In situ hybridizations data (Allen brain, our oligo probe data and unpublished FISH data) clearly show that expression is low in CA1 neurons. Functional analyses of shisa9 KO mice are also consistent with very low expression in CA1 neurons and high expression in DG granule cells (von Engelhardt et al., Science 2010, Khodosevich et al., Neuron 2014). Because I am not an expert on single cell RNAseq analyses, I asked a colleague who is an expert in this kind of analyses about his opinion of the discrepancy of the Sten Linnarsson’s data and other expression analyses. His answer: “Mouse brain atlas (based on Sten Linnarsson’s data) is a good resource for any initial reference. However, data might be different - they captured only ~800 genes/cell due to shallow sequencing of so many cells. So, there might be some discrepancies with ISH/IHC”. He also did some analysis of his own data with deeper sequencing: “We re-analyzed their data in the OB and these were not really overlapping with our data”. Due to these discrepancies I am somewhat hesitant to cite the Linnarsson paper as a reference for expression of shisas/CKAMPs. However, if the reviewer thinks that this is needed, I am happy to cite the paper. I also rephrased the statements about high expression in “dentate gyrus granule cells” to high expression in the “dentate gyrus” (which would include interneurons that may express shisa9 at higher levels). Similarly, I changed from “periglomerular cells” to “glomerular layer”, which would include the olfactory bulb neuroblasts that show in Linnarssons paper a very high expression. 

8. The author states that: “The uniqueness of the expression profiles with comparably little overlap suggests that a given neuron type expresses only one or two CKAMP family members. This is an observation that has been made for other AMPA receptor interacting proteins as well [25]”. There is to my knowledge no convincing data that brings evidence for the separate expression or the colocalization of the SHISA/CKAMP proteins.  The author should be clear about this.

It is true that many neuron types could express several SHISA/CKAMP mRNAs at certain levels. However, the in situ hybridization data shown in Figure 2 (but also from Allen brain atlas) show that the mRNA expression patterns of the different SHISA/CKAMP mRNAs are quite distinct. I do think that it is correct to say (and informative for the reader) that these data suggestthat most neurons express at high levels only one or two SHISA/CKAMP mRNAs. I rephrased the sentence accordingly (line 192-193).

Minor comments

Figure 2. It is a pity that the author is using this low resolution in situ hybridization data. This does not do justice to the information from better types of expression data in Allen Brain Atlas and the recent single cell sequencing data for mouse brain. The latter data also provide a much better insight into expression of the SHISA/CKAMP family than the in situ hybridisations. If the in situ hybridization of the author is kept, than please use better resolution (as the author did previously) and indicate the structures in the image (as the author did previously).

I agree that the in situs had a too low resolution and now show in situ hybridization data with higher resolution. 

Line 61: “Indeed, CKAMP44 variants with individual cysteines mutated to alanine could notmodulate AMPA receptor function.”  I have the impression the author wants to say the reverse “…..individual cysteines mutated to alanine could modulate AMPA receptor function.

I rephrased the sentence.

Line 73: “These data showed that this domain is responsible for the interaction with AMPA receptors.” Deletion of 20 amino acids is quite substantial and may well affect tertiary structure or protein topology by which interaction with the AMPAR is lost.  This alternative should be mentioned.

That is of course true and I mentioned the alternative explanation (line 111-115)

By the way, we tested two other mutants in which only 6 amino acids (180-185, and 186-191) of the 20 were deleted. Both mutants did not bind to GluA1, narrowing the potential interaction site even more. However, the same alternative explanation for a loss of interaction that the reviewer mentions holds true for these mutants as well. 

Line 136: “Interestingly, CKAMP52-containing AMPA receptors do not contain CKAMP44 despite the fact that both proteins are expressed in hippocampal neurons.” I think the author needs to report that SHISA9/CKAMP44 is most abundantly expressed in inhibitory neurons in the brain, whereas SHISA6/CKAMP52 is most abundantly expressed in excitatory cells. Again, please refer to novel insight from single cell RNA sequencing (see above, PMID:25700174).

See also the response to comment No. 7. 

Functional data, and in situ hybridization data clearly show that Shisa9/CKAMP44 is expressed in dentate gyrus granule cells (von Engelhardt et al., Science 2010; Khodosevich et al., Neuron 2014; Allen brain Atlas). It is of course possible that shisa9/CKAMP44 is expressed at even higher levels in some interneurons. However, there is certainly expression of shisa9 and shisa6 in the same cell type (not necessarily in each individual cell of course and most likely at different levels). I still think that the observation that “CKAMP52-containing AMPA receptors do not contain CKAMP44” is interesting and is worth mentioning. 

Figure 3. There is something wrong with the spacing of the lettering in the figure. They space very unequally. Please repair.

Spacing looks fine in our Figure 3. To the editor: Could it be that different pdf readers distort the letter spacing? 

Line 330: “In  contrast, the reduction in receptor diffusion cannot be the relevant mechanism by which CKAMP52 increases facilitation.” This is true, but Klaassen et al made the case that the stabilized receptors are less sensitive to desensitization, which would decrease the need for lateral diffusion. It would be good to sketch theis entire line of reasoning here.

I added the entire line of reasoning (line 405-407).

Line 383: ‘auxiliary’ is misspelled.

I have corrected this typo.

Round 2

Reviewer 2 Report

Please find two remaining comments attached.

Author Response

I changed the text according to the suggestion of the reviewer